# More Than Routing: Joint GPS and Route Modeling for Refine Trajectory Representation Learning

## ABSTRACT

Trajectory representation learning plays a pivotal role in supporting various downstream tasks. Traditional methods in order to filter the noise in GPS trajectories tend to focus on routing-based methods used to simplify the trajectories. However, this approach ignores the motion details contained in the GPS data, limiting the representation capability of trajectory representation learning. To fill this gap, we propose a novel representation learning framework that Joint GPS and Route Modelling based on self-supervised technology, namely **JGRM**. We consider GPS trajectory and route as the two modes of a single movement observation and fuse information through inter-modal information interaction. Specifically, we develop two encoders, each tailored to capture representations of route and GPS trajectories respectively. The representations from the two modalities are fed into a shared transformer for inter-modal information interaction. Eventually, we design three self-supervised tasks to train the model. We validate the effectiveness of the proposed method on two real datasets based on extensive experiments. The experimental results demonstrate that JGRM outperforms existing methods in both road segment representation and trajectory representation tasks. Our source code is available at https://anonymous.4open.science/r/JGRM-DAD6/.

## KEYWORDS

Trajectory representation learning, Spatio-temporal data mining, self-supervised learning

## 1 INTRODUCTION

With the development of location-based services including map services and location-based social networks, the generation and analysis of trajectory data have become pervasive, providing valuable insights into the mobility of various entities, such as individuals, vehicles and animals. These trajectory data contain rich spatial and temporal information that can be applied to urban planning [2, 14], urban emergency management [16, 37], infectious disease prevention and control [1, 10], and intelligent logistics systems [11, 24, 27]. However, to exploit the full potential of these data, the development of effective trajectory representation methods has emerged as a critical topic. Trajectory representation learning focuses on transforming raw trajectory data into meaningful and compact representations that can be used for a variety of tasks, such as travel time estimation [29], trajectory classification [22] and Top-k similar trajectory query [33].

Early studies on learning trajectory representations were based on sequential models designed for a specific downstream task and trained using the specific task loss [23, 28, 34]. These representations are not generalized and tend to crash on other tasks. To solve this problem, seq2seq-based methods have been proposed, which are trained by reconstructive loss [8, 21, 35] to make generalized representations. After that, due to redundancy and noise in the GPS

trajectory, the method using route trajectory instead of raw GPS trajectory became mainstream. These methods introduce many NLP techniques, including Word2Vec and BERT, due to the similarity between route trajectories and natural language sentences [5, 32]. Recently, with the rise of graph neural networks, researchers have begun to focus on the spatial relationships between road segments. Therefore, some two-step methods [9, 12] have been proposed, which first model the spatial relationships between road segments using the topology of the road network, and then use the updated road segments for temporal modeling using the sequence model. On this basis, a multitude of self-supervised training methods have been designed in order to train trajectory representation models in a task-free manner [17, 25, 31].

However, these methods simply treat road segments as conceptual entities (similar to words in natural language), ignoring the fact that a road segment is a real geographic entity that can interact with objects that pass through it. For example, when a road segment is congested, the movement pattern of passing vehicles is different than when the road is clear. So, different types of roads and different traffic states can really affect mobility. To this end, we believe that modeling road segments as geographic entities can effectively improve trajectory representation. Fortunately, the raw GPS points can serve as localized observations of the geographic entity. However, while the GPS trajectory contains richer information, it also contains a large amount of redundancy and noise and is not effective at capturing high-level transfer patterns. An intuitive idea is to combine the GPS view and the route view together to represent the trajectory more comprehensively.

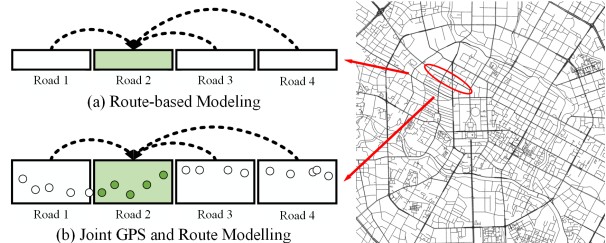

**Figure 1: Route Modeling v.s. Fusion Modeling.**

As shown in Figure 1(a), a road segment in the route trajectory, can only be modeled through preceding and succeeding road segments and lack of direct self-observation. In contrast, road segments in GPS trajectories offer much richer sampling information allowing for a fine-grained representation of road segment entities. Moreover, the context of road segments in the route trajectory can further refine the road representations. In fact, GPS trajectory and route trajectory simultaneously describe different perspectives of the same movement behavior and can complement each other. The GPS trajectory describes the movement details of the object, which can reflect the interaction of the object with the geospatial space as it moves, and can better model the road segment entities. However,

GPS trajectories are inherently noisy and redundant, which can degrade performance when modeling sequences. Route trajectory describes the travel semantics of an object, has a robust state transfer record, and can reflect travel intentions and preferences. What's more, it loses movement details and cannot effectively model states in geospatial space. Therefore, joint modeling route trajectory and GPS trajectory can realize the effective combination of macro and micro perspectives.

In practice, joint modeling two types of trajectories is a non-trivial task: *(1) Uncertainty in GPS Trajectory.* There are a large number of redundant and noisy signals in the GPS trajectory, and they can seriously affect the computational efficiency and performance of the model. *(2) Spatio-temporal Correlation in Route Trajectory.* Route has a complex spatio-temporal correlation, the topology of the road network must be taken into account when an object undergoes a road segment transposition, and the travel time of a road segment is related to the historical traffic pattern and the current travel state. *(3) Complexity of Information Fusion.* Consider that although GPS trajectory and route trajectory describe the same concept, the two data sources imply two domains due to different perspectives. Fusing information from different domains is a challenge. Furthermore, in order to obtain the generalized trajectory representation, we would like to train the model using the self-supervised paradigm.

To address these problems, we develop a novel representation learning framework that joint GPS and route modeling based on self-supervised technology, namely JGRM. It contains three components, the GPS encoder, the route encoder, and the modal interactor, which correspond to the three challenges above. Specifically, The GPS encoder uses a hierarchical design to solve the redundancy and noise problems in GPS trajectories by embedding the corresponding sub-trajectories through the road segment grouping bootstrap. The route encoder uses a road network-based spatial encoder GAT and a lightweight temporal encoder TE, to capture the spatio-temporal correlation in the route trajectory. Autocorrelation of the route trajectory is captured by the Transformer in the route encoder. Finally, we treat the two trajectories as two modalities and use the shared transformer as a modal interactor for information fusion. We also designed two self-supervised tasks to train our JGRM, which are MLM and Match. The MLM obtains supervisory information by recovering road segments that were randomly masked before the trajectory was fed into the encoder. In contrast, Match exploits the fact that the GPS trajectory and the route trajectory are paired to generate pairwise losses to guide the two modalities to align the representation space before it is fed into the modal interactor.

Our contributions are summarized as follows:
- To the best of our knowledge, we are the first to propose joint modeling of GPS trajectories and route trajectories for trajectory representation learning.
- We propose a trajectory representation learning framework based on the idea of multimodal fusion which is named JGRM. It consists of a hierarchical GPS encoder to model the characteristics of road entities, a route encoder that considers the spatio-temporal correlation of trajectories, and a modal interactor for information fusion.
- Two self-supervised tasks are designed for model training, which are generalizable to subsequent research works. Among them,

MLM is used to reconstruct the spatio-temporal continuousness of the trajectory itself, and CMM is used to fuse mobility information from different views.
- Extensive experiments on two real-world datasets validate that JGRM achieves the best performance across various settings.

## 2 RELATED WORK

In our work, the trajectory represents a set of raw GPS sampling points and corresponding timestamps, which constitute a unique record, and the route is a sequence of roads through which a trajectory passes Both GPS and route describe the behavior of the trajectory from different perspectives. We review related work on modeling representations using these components in the following.

### 2.1 Route Trajectory Representation Learning

Route represents the approximate travel path of trajectory and can describe the structure of the road network, meanwhile, route trajectory representation learning can learn the knowledge information of trajectory and road network. Recent studies have found that route trajectory representation learning can quickly discover the travel semantics of an object, reflect the travel intentions and preferences of an object through the route context, and better model movement trajectories.PIM [31] uses node2vec to learn road representations and uses a mutual information-maximizing RNN to generate trajectory representations, but completely ignores the entire road network. While Toast [5] learns the node representation by adopting the skip-gram word2vec model and then uses a transformer to perform sequence modeling on roads traversed by a route, which learns road networks and trajectories separately but does not model them well in a fused way. At the same time, some studies have focused on temporal information in routes with underlying semantics. Trembr [12] is the first to consider adding timestamps to each road segment during decoding.START [17] captures the urban periodicity model that is not present in Trembr and the irregular time intervals between trajectory samples representing the semantics are richer.JCLRNT [25] and START unite information about road networks and routes, but lack the fine-grained information of GPS so that they perform poorly on some trajectory tasks.

Since the smallest granularity of these methods is the road, these models emphasized the learning of inter-road transitions but were unable to capture the intra-road moving details. Thus it is not possible to effectively model the state of geospatial.

### 2.2 GPS Trajectory Representation Learning

In comparison to route representation methods, trajectory representation emphasizes the learning of intra-trajectory features, such as the change of average speed, change of speed, and the change of rate of turn, etc. The richer sampling information from GPS enables fine-grained modeling of road segment information, and the combination of contextual road segments can also help to better understand road segments. Two of the earliest works that introduce the concept of trajectory representation learning are Traj2vec [35] and t2vec [21], which focus on the use of GPS data to reconstruct learning trajectories. Sometimes GPS's detailed spatial information supports its trajectory representation learning approach for more specific downstream tasks. NeuTraj [33] and CL-TSim [6] aim to

learn trajectory representations for approximate trajectory similarity computation, which uses grid cells to encode spatial latent representations of GPS points. And Trajformer [22] inspired by Transformer's trajectory sequence processing, combines GPS point input features and spatio-temporal interval information to generate continuous point embeddings to accelerate the representation learning and is used for the classification task of trajectories.

However, the GPS trajectory contains a large amount of redundant information and noise that does not effectively capture the transfer patterns at the trajectory level, so using only GPS trajectory representations is insufficient.

Joint modeling GPS trajectory and route trajectory is a research gap compared to previous work.

## 3 OVERVIEW

### 3.1 Preliminaries

DEFINITION 1. *(Trajectory) A trajectory $\tau_i$ represents the change in the position of an object over time. In this paper, a trajectory is observed from the GPS view and the route view, denoted as $g_i$ and $r_i$, respectively.*

DEFINITION 2. *(GPS Trajectory) GPS trajectory is a sequence of GPS points, denoted as $g_i =< gp_1, gp_2, \ldots, gp_n >$, each point $gp_i = (lat_i, lng_i, t_i)$ containing latitude, longitude and timestamp. $x_{\tau_i}^G$ denotes the GPS view feature of trajectory $\tau_i$.*

DEFINITION 3. *(Road Network) A road network is denoted as a directed graph $G = (V, E, A)$, where $V = \{v_1, v_2, \ldots, v_{|V|}\}$ is the set of vertices, each vertex $v_i$ refers to a road segment. $E \subseteq V \times V$ is the set of directed edges, each edge $e_{ij} =< v_i, v_j >$ refers to a intersection between road $v_i$ and $v_j$. $A \in \mathbb{R}^{|V| \times |V|}$ is a binary adjacency matrix of the road network $G$ that describes whether there are directed edges between two road segments.*

DEFINITION 4. *(Route Trajectory) Route Trajectory is a chronological sequence of visited record $r_i =< rp_1, rp_2, \ldots, rp_m >$, with each record $rp_j = (v_j, t_j)$ containing the road ID and the corresponding timestamp. $x_{\tau_i}^R$ denotes the route view feature of trajectory $\tau_i$.*

### 3.2 Problem Statement

For a trajectory $\tau_i$, given the GPS view feature $x_{\tau_i}^G$ and the route view feature $x_{\tau_i}^R$, our goal is to obtain a d-dimensional generalized representation of the trajectory $z_{\tau_i}$ and the road segments $\{z_{v_j}, v_j \in V_{\tau_i}\}$ appearing in the trajectory $\tau_i$, respectively.

### 3.3 Framework Overview

The framework of JGRM is shown in Figure 2, which consists of three modules to obtain a d-dimensional representation for each trajectory and road segment contained therein:

- **GPS Encoder**, which first encodes the road segments using the GPS sub-trajectories of the corresponding road segments, then refines the road segment representations through the sequential relationship between them to obtain the GPS view representations of the trajectory and the road segments contained therein.
- **Route Encoder**, which encodes the topological relationship between road segments and the temporal context separately, and

fuses the spatio-temporal context encoding with the road segment embeddings to obtain the road segment representations. These road segment representations are refined using sequential correlation, which ultimately yields the route view representations of the trajectory and the road segments in the trajectory.
- **Modal Interactor**, which further enhances the output representation with the interaction between two views so that an ideal representation can be achieved by fully integrating knowledge from road entity and trajectory information.

The whole framework is trained by the self-supervised paradigm, which includes two types of tasks, MLM (Mask Language Modeling) [7] and CMM (Cross-Modal Matching) [15]. The MLM task randomly masks some road segments before the trajectories are fed into the GPS and Route Encoder, and eventually rebuilds these road segments using the output of the modal interactor. The reconstruction error is used as a supervised signal to train the model. Note that the GPS and Route views are masked by the same road segments, hence the term Shared Mask. The CMM task refers to the fact that the trajectory representations of different views corresponding to the same trajectory are supposed to be paired, so the matched result of trajectory representations can be utilized to provide self-supervised signals, which are outputted by two encoders. Overall, the model is supervised by three losses, the GPS MLM loss, the Route MLM loss, and the GPS-Route Match loss.

## 4 METHODOLOGY

In this section, we first introduce three modules of JGRM in detail and then illustrate how the self-supervised tasks help to train the model.

### 4.1 GPS Encoder

The GPS encoder aims to encode the GPS trajectory to obtain the trajectory representation and the corresponding road segment representation in an efficient and robust manner.

**Main Idea.** Modeling GPS trajectories as traditional sequence data would focus too much on the endpoints of the trajectory, making it inefficient to accommodate long trajectories. In addition, there is noise and redundancy in the GPS trajectory that affects the sequence model representation capability. Considering these issues, we propose to model the road segments in the GPS trajectory individually and refine these road segment representations through sequence dependency. The reason for this is that modeling road segments individually ensures the independence of road segments as geographic entities, regardless of trajectory length. And sequence-based refinement can smooth the noise and redundant signal in each road segment. A hierarchical bidirectional GRU is designed to implement this two-stage modeling. A hierarchical bidirectional GRU is designed to implement this two-stage modeling, which consists of intra-road BiGRU and inter-road BiGRU. Similar to the previous presentation, intra-road BiGRU is used to encode segment entities and inter-road BiGRU refines the segment representation obtained from the former.

**Implementation.** To implement the above idea, we first use the map-matching algorithm to obtain the correspondence between sub-trajectories and road segments. An assignment matrix $B_{\tau_i}$ is created when the raw GPS trajectory is transformed into a route

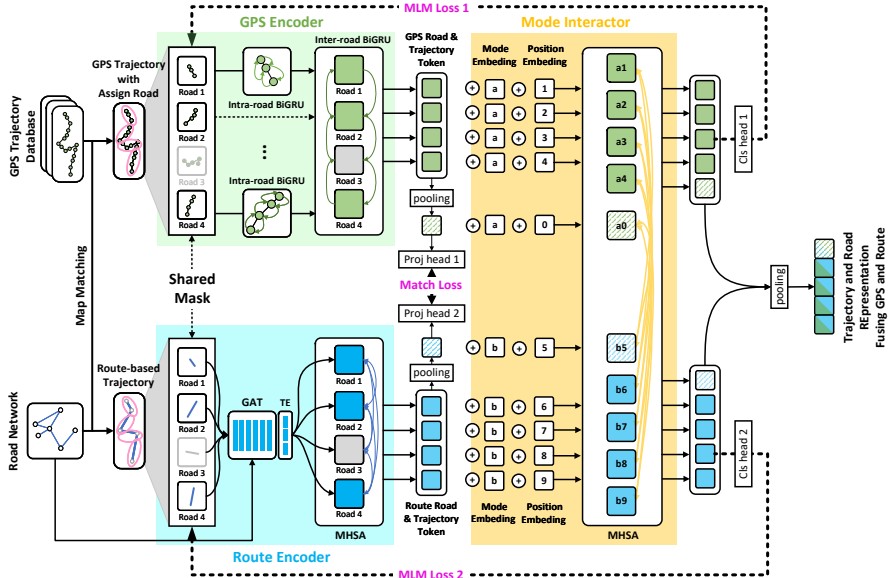

**Figure 2: The Framework of JGRM.**

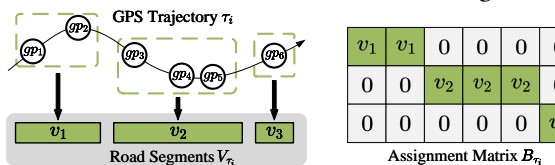

**Figure 3: An example of an assignment matrix.**

trajectory by the map matching algorithm. It describes the mapping of raw GPS points to road segments, as shown in figure 3. The i-th row of the assignment matrix indicates that the i-th sub-trajectory corresponds to road segment $v$.

Then, for each GPS trajectory, we first extract 7 features in each GPS point that describe the kinematic information of the trajectory: longitude, latitude, speed, acceleration, angle delta, time delta, and distance. $x_{\tau_i}^G \in \mathbb{R}^{n_i \times 7}$ indicates the feature matrix of GPS trajectory $\tau_i$, where $n_i$ is the trajectory length. Before the data is fed into the intra-road BiGRU, we need to organize the original feature matrix $x_{\tau_i}^G$ according to sub-trajectories. The records of sub-trajectories are maintained in the assignment matrix. Each sub-trajectory is expressed as follows:

$$I_{s_j}, v_j = B_{\tau_i}[j]$$
$$x_{s_j}^G = x_{\tau_i}^G \left[ I_{s_j} \right] \tag{1}$$

where $s_j = \left\{ gp_k, gp_{k+1}, \ldots, gp_{k+l_j-1} \right\}$ denotes the j-th subtrajectory in the assignment matrix. $I_{s_j} = \left[ k, k+1, \ldots, k+l_j-1 \right]$ is the set of indexes for each GPS point in the sub-trajectory $s_j$. $v_j \in V_{\tau_i}$ denotes the road segment and $x_{s_j}^G$ is feature matrix corresponding to the sub-trajectory $v_j$. $l_j$ is the length of the sub-trajectory. Next, the feature matric $x_{s_j}^G$ is fed into the intra-road BiGRU to get sub-trajectory hidden representation:

$$\overrightarrow{h_{s_j}^G}, \overleftarrow{h_{s_j}^G} = BiGRU_{intra}(x_{s_j}^G) \tag{2}$$

where $\overrightarrow{h_{s_j}^G}, \overleftarrow{h_{s_j}^G} \in \mathbb{R}^{l_j \times d_{intra}}$ are the forward and backward hidden representations of the sub-trajectory, respectively. The outputs of the intra-road BiGRU are sent to inter-road BiGRU to obtain the compressed road segment representation using sequence information.

$$\overrightarrow{H_{V_{\tau_i}}^G}, \overleftarrow{H_{V_{\tau_i}}^G} = BiGRU_{inter}(\left[\overrightarrow{h_{s_0}^G}(l_0-1), \overrightarrow{h_{s_1}^G}(l_1-1), \ldots, \overrightarrow{h_{s_{m_{i-1}}}^G}(l_{m_{i-1}}-1)\right]) \tag{3}$$

where $\overrightarrow{H_{V_{\tau_i}}^G}$ and $\overleftarrow{H_{V_{\tau_i}}^G}$ represent the set of all forward and backward road segment representations in the GPS trajectory $\tau_i$, respectively. $m_i$ is the number of sub-trajectories in the GPS trajectory $\tau_i$. The final road segment representation is obtained by concatenating them, denoted by $Z_{V_{\tau_i}}^G = \left[ \overrightarrow{H_{V_{\tau_i}}^G}, \overleftarrow{H_{V_{\tau_i}}^G} \right], Z_{V_{\tau_i}}^G \in \mathbb{R}^{m_i \times 2d_{inter}}$. These road segment representations in the GPS trajectory are all sent to the mode interactor. And, we use a simple additive model to compute the trajectory representation:

$$z_{\tau_i}^G = \text{MeanPool}(\{z_{v_j}^G, v_j \in V_{\tau_i}\}) \tag{4}$$

where $z_{\tau_i}^G \in \mathbb{R}^{1 \times 2d_{inter}}$ is the representation vector of the trajectory $\tau_i$ in GPS view.

## 4.2 Route Encoder

In this module, we model route trajectory from the spatial and temporal perspectives. Eventually, road segment representations and trajectory representations of trajectory $\tau_i$ are obtained in the route view.

**Main Idea.** Route trajectory generation is constrained by the topology of the road network and the current traffic situation. Adjacency in the road network requires that adjacent segments are connected in the routing trajectory. Current traffic conditions then affect the driver's route planning, which is reflected in the probability of choosing each road because drivers tend to favor less

congested routes. To simulate this process, we propose to first use GAT to update the embedding of the road segment when new trajectories are observed in a streaming fashion. This design ensures that our model is updated to the road segment representation in real time with the observed trajectory data. From the temporal perspective, we propose to encode the time information using the contextual time and the actual travel times of each segment in the route trajectory. The context time describes the periodicity in the traffic flow, while the actual elapsed time further captures the current state of the road segment. Similarly, road segment representations that incorporate temporal and spatial information are finally refined based on the autocorrelation of the sequences. Given the complex dependencies between road segments, the transformer is the ideal choice to update road segment representations.

**Implementation.** We consider four types of features for encoding the route trajectory, including the road ID, the time delta, the minutes index (0-1439), and the day of the week index (0-6) of the start time. $x_{\tau_i}^R \in \mathbb{R}^{m_i \times 4}$ denotes the route feature matrix of the trajectory $\tau_i$, where $m_i$ indicates the number of road segments contained in the route trajectory.

Based on the above, the road embedding is first updated using the topology of the road network:

$$RE' = GATLayer(RE(V), A) \tag{5}$$

where $RE$ is road network embedding, which converts road IDs into dense vectors. $RE'$ is the road network embedding updated with the message passing in the graph. To encode the temporal information of the road segments, the two context times are embedded as discrete values, similar to road network embedding. And the actual travel time is embedded as a continuous value. In practice, inspired by the idea of binning, we maintain a conceptual embedding containing 100 virtual buckets. When the travel time is input, it is first transformed into a 100-dimensional vector, and the weights of each concept bucket are obtained by softmax. As a result, the travel time is transformed into a dense vector. This process is formulated as follows:

$$IE(\Delta t_j) = \text{Softmax}(FFN(\Delta t_j)) * W_{TE}$$
$$h_{v_j}^R = GE'(v_j) + TE_{min}(t_j) + TE_{week}(t_j) + IE(\Delta t_j) \tag{6}$$

$h_{v_j}^R$ is the representation of the segment $v_j$. $TE_{min}$ and $TE_{week}$ are the minute embedding and day-of-week embedding of the start time, respectively, and $IE$ is the travel time embedding. $W_{TE}$ is a learnable parameter matrix. Next, the updated road segment representations are fed into the Transformer encoder for refinement. For simplicity, no additional positional embedding is designed because the order is already included in the time coding.

$$H_{V_{\tau_i}}^R = [h_{v_0}^R, h_{v_1}^R, \ldots, h_{v_{m_i-1}}^R]$$
$$Z_{V_{\tau_i}}^R = \text{TransEncoder}(FFN(H_{V_{\tau_i}}^R)) \tag{7}$$

where $Z_{V_{\tau_i}}^R \in \mathbb{R}^{m_i \times d_{rep}}$ refer to the set of segment representations. At last, the average pooling is employed to obtain the trajectory representation:

$$z_{\tau_i}^R = \text{MeanPool}(\{z_{v_j}^R, v_j \in V_{\tau_i}\}) \tag{8}$$

where $z_{\tau_i}^R \in \mathbb{R}^{1 \times d_{rep}}$ is the representation vector of the trajectory $\tau_i$ in route view.

## 4.3 Mode Interactor

GPS trajectory and route trajectory can be treated as two observations of the same concept, similar to the two modal data. Inspired by multimodal pre-training studies [4, 18, 36], we introduce a shared transformer for cross-modal information interaction. For each modality, the input token undergoes modal embedding and positional embedding, respectively, to preserve modal identity:

$$e = z + ME(z) + PE(z) \tag{9}$$

We then feed these processed road segment representations and trajectory representations into the transformer encoder. The data are organized in the order of trajectory representation and road segment representation in both modalities:

$$[z_{\tau_i}^{G'}, Z_{V_{\tau_i}}^{G}{}', z_{\tau_i}^{R'}, Z_{V_{\tau_i}}^{R}{}'] = \text{TransEncoder}(FFN([e_{\tau_i}^G, E_{V_{\tau_i}}^G, e_{\tau_i}^R, E_{V_{\tau_i}}^R])) \tag{10}$$

where $z_{\tau_i}^G$ and $Z_{V_{\tau_i}}^G$ denote the trajectory representations and the set of road segment representations in the GPS view, $z_{\tau_i}^R$ and $Z_{V_{\tau_i}}^R$ and d ditto in the route view. Trajectory and road segment representations are calculated as the mean of two types of representations:

$$\hat{Z}_{V_{\tau_i}} = \text{MeanPool}([Z_{V_{\tau_i}}^{G}{}', Z_{V_{\tau_i}}^{R}{}']), \hat{Z}_{V_{\tau_i}} \in \mathbb{R}^{m_i \times d_{out}}$$
$$\hat{z}_{\tau_i} = \text{MeanPool}([z_{\tau_i}^{G'}, z_{\tau_i}^{R'}]), \hat{z}_{\tau_i} \in \mathbb{R}^{1 \times d_{out}} \tag{11}$$

## 4.4 Self-supervised Training

In order to obtain the generalized trajectory representation, we design two types of self-supervised tasks for training the proposed JGRM.

**MLM Loss.** MLM has been shown to perform well in self-supervised training on sequence data [7]. However, the road segments in the trajectory are constrained by the road network. Recovering these randomly masked independent tokens is relatively simple and insufficient to adequately train the model. To increase the difficulty of the task, we randomly mask the subpaths of length $l$ with probability $p$, where $l \geq 2$. To prevent the two types of trajectories from leaking to each other, a shared mask is executed on both the GPS trajectory and the route trajectory. Specifically, the shared mask hides the same road segments in both types of trajectories. Our task is to recover these segments using the corresponding token representations output by the mode interactor. The self-supervised task is trained by cross-entropy loss.

In practice, we first transform these token representations using the classification head. A layer feed-forward neural network is used as the classification head, which is different for GPS trajectory and route trajectory:

$$\tilde{z}_{v_j}^G = FFN_{gcls}(z_{v_j}^{G'}), \tilde{z}_{v_j}^R = FFN_{rcls}(z_{v_j}^{R'}) \tag{12}$$

where $\tilde{z}_{v_j}^G, \tilde{z}_{v_j}^R \in \mathcal{R}^{1 \times |V|}$ are the corresponding token vectors for the GPS view and the route view. The transformed vectors are then used to calculate the loss:

$$\mathcal{L}_T^{GMLM} = -\frac{1}{|T|} \sum_{\tau_i \in T} \frac{1}{|\mathcal{M}_{\tau_i}|} \sum_{v_i \in \mathcal{M}} \log \frac{\exp(\tilde{z}_{v_i}^G)}{\sum_{v_j \in V_{\tau_i}} \exp(\tilde{z}_{v_j}^G)}$$

$$\mathcal{L}_T^{RMLM} = -\frac{1}{|T|} \sum_{\tau_i \in T} \frac{1}{|\mathcal{M}_{\tau_i}|} \sum_{v_i \in \mathcal{M}} \log \frac{\exp(\tilde{z}_{v_i}^R)}{\sum_{v_j \in V_{\tau_i}} \exp(\tilde{z}_{v_j}^R)} \tag{13}$$

where $T$ is the set of trajectories and $\mathcal{M}$ is the set of masked segments in all trajectories. $\mathcal{M}_{\tau_i}$ is the set of masked segments in a given trajectory $\tau_i$.

**Match Loss.** The matching task is designed to guide the alignment of the two representation spaces, which are maintained by two encoders. Considering that the GPS trajectory and the route trajectory appear in pairs and can be referred from each other, we borrow design ideas from cross-modal retrieval studies [20]. For a trajectory set $T$, two types of trajectories can be retrieved from each other to generate $|T|^2$ match results. Each match result is a binary classification problem that can be optimized by cross entropy loss.

First the trajectory representations of the two encoder outputs are fed into the corresponding projection head for transformation.

$$\acute{z}_{\tau_i}^G = FFN_{porj1}(z_{\tau_i}^G), \acute{z}_{\tau_i}^R = FFN_{porj2}(z_{\tau_i}^R) \tag{14}$$

$\acute{z}_{\tau_i}^G, \acute{z}_{\tau_i}^R \in \mathcal{R}^{1 \times d_{proj}}$ are the vectors obtained after projection. We use a single fully-connected layer to discriminate the results of a single retrieval:

$$\hat{y}^{GR} = FFN_{pcls}([\acute{z}_{\tau_i}^G, \acute{z}_{\tau_i}^R]) \tag{15}$$

where $z_{\tau_i}^G$ and $z_{\tau_i}^R$ are the trajectory representations of the two encoder outputs. $\hat{y}^{GR}$ is the predict result. In practice, to overcome sparse supervision and computational efficiency, we replace the above loss with a simpler form. For each pair, only 3 loss terms are considered, which are the match results of the two corresponding positive samples, the positive GPS sample and the negative route sample, and the negative GPS sample and the positive route sample. This design based on triplet loss can effectively improve training efficiency. Note that We use only the one that most closely resembles the current query trajectory as the negative sample in each retrieval. The formula is the following:

$$\mathcal{L}_T^{\text{Match}} = \frac{1}{3}[CE(\hat{y}^{GR}, y^{GR}) + CE(\hat{y}^{G\bar{R}}, y^{G\bar{R}}) + CE(\hat{y}^{\bar{G}R}, y^{\bar{G}R}]$$

$$CE(\hat{y}, y) = -\frac{1}{|T|} \sum_{\tau_i \in T} y_{\tau_i} \log(\hat{y}_{\tau_i}) \tag{16}$$

where $\bar{G}$ and $\bar{R}$ denote negative samples at GPS view and route view. $y^{GR}, y^{G\bar{R}}$ and $y^{G\bar{R}}$ are 1,0,0, respectively. The overall loss is defined as:

$$\mathcal{L}_T = w_1 \mathcal{L}_T^{GMLM} + w_2 \mathcal{L}_T^{RMLM} + w_3 \mathcal{L}_T^{Match} \tag{17}$$

where $w_1$, $w_2$, and $w_3$ are the hyperparameters to balance the three tasks.

# 5 EXPERIMENTS

## 5.1 Experimental Settings

In this section, we evaluate the performance of JGRM on a series of experiments in two real-world datasets, which are summarized to answer the following questions:

- **RQ1**: How does JGRM's performance compare to other comparison methods on four downstream tasks?
- **RQ2**: How does every module that we design contribute to the model performance?
- **RQ3**: How effective are our pre-trained models?

**Dataset Description**. We evaluate our approach in two real-world datasets, which are Chengdu and Xi'an. Each of these includes GPS trajectories, route trajectories, and road networks. GPS trajectories are obtained from public datasets released by Didi Chuxing [1]. Corresponding road networks are collected from OSMNX [3]. The road network data includes the road type, road length, number of lanes, and topological relationships. We use only the topological relationships of the road segments during training, which is different from some baselines. The raw GPS trajectories are mapped into the road network using the map matching algorithm [30] to obtain the route trajectories and assignment matrix. The assignment matrix indicates the mapping of GPS sub-trajectories to road segments. To be fair, we filtered out the road segments that were not covered by trajectories. Similarly, we remove trajectories with fewer than 10 road segments, which would affect model performance. Both datasets have the same time span, which is 15 days. We divide the data from the first 13 days as the training set, the 14th day as the validation set, and the 15th day as the testing set. The details of each dataset are summarized in Table 3.

**Downstream Tasks and Metrics**. We use similar experimental settings in [5, 25]. A total of four tasks were used to evaluate the model performance, including two segment-level tasks and two trajectory-level tasks. Segment-level tasks consist of road classification and road speed estimation, where the former is a classification task and the latter is a regression task. They are used to evaluate the characterization capabilities of road segment representations across tasks with different granularity. In these tasks, the representations of the same road segments in different trajectories are averaged as static representations, that is the input data. The trajectory-level tasks include travel time estimation and top-k similarity trajectory query, which evaluate trajectory representations at different semantic levels. The travel time estimation stands for the shallow-order semantics of trajectory and is related to the spatio-temporal context and the current traffic state. Top-k similar trajectory query is more related to OD (Origin-Destination) pair and driving preferences and belongs to higher-order semantics.

Note that we fixed the model parameters and only trained classification or regression heads during the evaluation. In the top-k trajectory similarity query task, we directly use the raw output of the model as trajectory representations without finetune. The experimental setup for these four tasks is shown in the Appendix.

---

[1]https://outreach.didichuxing.com/

## 5.2 Performance Comparison (RQ1)

We compare our proposed JGRM with the following 10 methods that are categorized into 4 groups. To be fair, all of the above methods were trained to use 10w trajectories.

**Random Initialization.**

- **Embedding**: The road segment representation is randomly initialized.

**Graph-based Trajectory Representation Learning.**

- **Word2vec**[26]: It use the skip-gram model to obtain the road segment representation based on co-occurrence.
- **Node2vec**[13]: It efficiently learns embeddings for nodes in a network by sequences generated by random walks.
- **GAE**:[19]: It is a classical graph encoder-decoder model that learns the node embedding by reconstructing the adjacency matrix.

The trajectory representation of such methods is given by the average of the road segment representation.

**GPS-based Trajectory Representation Learning.**

- **Traj2vec**[35]: It converts raw GPS trajectory into feature sequence and adopts seq2seq model to learn the trajectory representation.

**Route-based Trajectory Representation Learning.**

- **Toast**[5]: Built upon the Skip-gram pretraining result for node embeddings, and uses them on the MLM task and trajectory discrimination task to train the model.
- **PIM**[31]: It employs contrastive learning on the samples generated by the shortest path, and their variations by swapping the nodes between positive and negative paths for road networks.
- **Trember**[12]: It first transforms trajectory into spatio-temporal sequences, then passes through RNN-based Traj2vec to obtain the trajectory representation.
- **START**[17]: The authors propose a trajectory encoder that integrates travel semantics with temporal continuity and two self-supervised tasks.
- **JCRLNT**:[25]: JCLRNT develops a graph encoder and a trajectory encoder to model the representation of road segment and trajectory, respectively. These representations were organized to train the model through three comparison tasks.

Tables 1 and 4 show the comparison results of all methods. We run all models with 5 different seeds and report the average performance. As can be observed, our JGRM achieved the best performance on all four downstream tasks for both real-world datasets. This demonstrates the effectiveness of JGRM in jointly modeling the GPS trajectory and route trajectory in a self-supervised manner. Also, for each task, we labeled the second and third-best methods († and ‡). JGRM obtained significant performance improvements in almost all metrics. In addition, we developed a larger version denoted as JGRM* trained using 50w trajectories, which achieves better performance.

Compared to other methods, our method performs much better than other baselines on the road segment level task. It suggests that effective modeling of road segments can significantly improve the performance of trajectory representation learning. Sequence models such as Toast, PIM, and Trember tend to perform better performance in trajectory-level tasks, indicating that modeling spatio-temporal

correlation in trajectory is necessary. Among them, the GPS-based representation learning method traj2vec performs poorly, which is due to the fact that it ignores the noise and redundancy in the GPS trajectory. Interestingly, the graph-based trajectory representation learning approach achieved unexpected results on the sequence-level task. This suggests that the topology between road segments is important for trajectory representation. Note that we did not use road attributes during training because it is very expensive to collect data accurately. It causes START to produce a significant performance degradation.

## 5.3 Ablation Study (RQ2)

To evaluate the effects of each module in JGRM, we performed ablation experiments on 8 variants: (1) **w/o MLM Loss**: This variant leaves the model structure unchanged and removes two MLM losses. (2) **w/o Match Loss**: Similar to the previous one, which only removes the Match loss. (3) **w/o GPS Branch**: This variant removes the GPS encoder and modal interactor and their corresponding loss functions, keeping only the route MLM loss. (4) **w/o Route Branch**: This variant is similar to the one above, only retains the GPS encoder and GPS MLM loss. (5) **w/o Time Info**: This variant masks the input temporal information. (6) **w/o Mode Interactor**: This variant only removes mode interactor. Two MLM losses are calculated using the outputs of encoders in this case. (7) **w/o GAT**: Remove the GAT from the model, and leave the others as they are. (8) **w/o Mode Emb**:This variant only remove the modal embedding.

The results of the ablation experiments in Chengdu are shown in Table 5. Due to space limitations, the Xi'an results are included in the appendix. We can observe that the overall performance of our method beats all variants. On part tasks, the variants outperform our approach, marked for *uparrow*. It shows that different modules focus differently on different types of tasks, our JGRM is a trade-off. MLM achieves the best performance improvement among all variants, indicating the effectiveness of the improved self-supervised task. Joint modeling also yielded significant improvements over methods that used only one type of trajectory modeling. Other modules work well for specific types of tasks. The combination of these modules can be customized to meet specific needs.

## 5.4 Pre-training Effect Study. (RQ3)

To explore the pre-training effects of the model, we report the travel time estimation results in both the re-training (No Pre-train) and the regression head fine-tuning (Pre-train). The results are presented in Figure 4. The pre-trained model shows different gains in the experiments of the two cities. Figure 4 shows that the pre-trained model has rich prior knowledge and can significantly reduce the amount of data required to train the model. Xian's experimental results show that the pre-trained model has the ability to prevent overfitting and can continuously improve the model performance with the increase of training data.

In Figure 5, we report the results of models trained on datasets of different sizes for road segment speed inference and similar trajectory query. The model shows better performance as the data size increases. We find that our proposed JGRM has a large model capacity that performance can be continuously improved with training.

**Table 1: Model comparison on four downstream tasks in Chengdu.**

| | Road Classification | | Road Speed Inference | | Travel Time Estimation | | Top-k Similar Trajectory Query | | |
|---|---|---|---|---|---|---|---|---|---|
| | Mi-F1 | Ma-F1 | MAE | RMSE | MAE | RMSE | MR | HR@10 | No Hit |
| Embedding | 0.3853 | 0.2757 | 3.561 | 4.6437 | 102.592 | 132.4559 | 9.4693 | 0.85 | 0 |
| Word2vec | 0.5514 | 0.5137 | 3.5004 | 4.5424 | 87.1612‡ | 115.6605‡ | 12.4355 | 0.7998 | 0 |
| Node2vec | 0.408 | 0.364 | 3.5761 | 4.6623 | 88.1243 | 117.3834 | 4.103† | 0.9127† | 0 |
| GAE | 0.4373 | 0.3805 | 3.287‡ | 4.2134‡ | 90.2352 | 122.9764 | 4.4584‡ | 0.9067‡ | 0 |
| Traj2vec | 0.4828 | 0.399 | 2.856† | 3.81† | 99.0706 | 128.4441 | 67.5899 | 0.55 | 839.2 |
| Toast | 0.6276† | 0.6195† | 3.3201 | 4.3777 | 86.0053† | 114.2109† | 5.9169 | 0.8696 | 0 |
| PIM | 0.4618 | 0.4457 | 3.4841 | 4.5737 | 87.6526 | 116.533 | 5.109 | 0.8902 | 0 |
| Trember | 0.611‡ | 0.6059‡ | 3.3955 | 4.447 | 90.9035 | 119.0926 | 17.9627 | 0.7427 | 0.1 |
| START | 0.409 | 0.3366 | 3.5269 | 4.6084 | 89.7182 | 117.9891 | 6.9448 | 0.909 | 30.7 |
| JCRLNT | 0.5169 | 0.466 | 3.441 | 4.5016 | 100.1113 | 129.591 | 20.0152 | 0.7323 | 0.6 |
| JGRM | **0.7198** | **0.7228** | **2.5783** | **3.5452** | **83.3306** | **110.7224** | **2.2111** | **0.9492** | 0 |
| JGRM* | **0.8067*** | **0.8111*** | **2.3162*** | **3.2953*** | **80.4002*** | **108.0134*** | **1.1363*** | **0.9735*** | 0 |
| improvement | 14.69% | 16.67% | 10.77% | 7.47% | 3.21% | 3.15% | 85.56% | 4% | / |
| improvement* | 28.54% | 30.93% | 23.31% | 15.62% | 6.97% | 5.74% | 261.08% | 6.66% | / |

**Table 2: Ablation experiment on four downstream tasks in Chengdu.**

| | Road Classification | | Road Speed Inference | | Travel Time Estimation | | Top-k Similar Trajectory Query | | |
|---|---|---|---|---|---|---|---|---|---|
| | Mi-F1 | Ma-F1 | MAE | RMSE | MAE | RMSE | MR | HR@10 | No Hit |
| JGRM | 0.7198 | 0.7228 | 2.5783 | 3.5452 | 83.3306 | 110.7224 | 2.2111 | 0.9492 | 0 |
| w/o MLM Loss | 0.5233 | 0.4804 | 3.4752 | 4.5521 | 122.7088 | 152.9668 | 26.4418 | 0.0085 | 4725.8 |
| w/o Match Loss | 0.7178 | 0.7232 ↑ | 2.6075 | 3.5947 | 82.5453 ↑ | 110.2262 ↑ | 2.3396 | 0.9441 | 0 |
| w/o GPS Branch | 0.6245 | 0.6206 | 3.2008 | 4.2258 | 83.6647 | 111.4075 | 1.6037 ↑ | 0.963 ↑ | 0 |
| w/o Route Branch | 0.6122 | 0.5929 | 2.8302 | 3.7668 | 95.2015 | 124.4988 | 9.2601 | 0.8381 | 0 |
| w/o Time Info | 0.7331 ↑ | 0.7361 ↑ | 2.6225 | 3.5866 | 84.1749 | 111.6983 | 5.6927 | 0.8745 | 0 |
| w/o Mode Interactor | 0.6043 | 0.5859 | 2.7381 | 3.7303 | 82.9407 ↑ | 110.4866 ↑ | 1.4601 ↑ | 0.965 ↑ | 0 |
| w/o GAT | 0.7173 | 0.7225 | 2.706 | 3.654 | 82.2657 ↑ | 110.038 ↑ | 1.1554 ↑ | 0.9732 ↑ | 0 |
| w/o Mode Emb | 0.7161 | 0.7222 | 2.7439 | 3.6944 | 83.8222 | 111.5119 | 2.535 | 0.9417 | 0 |

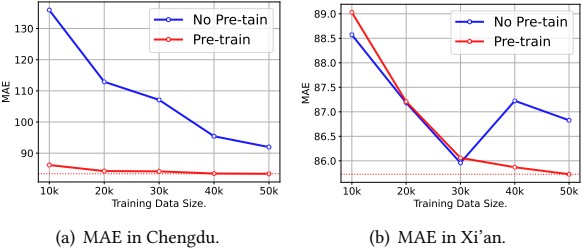

(a) MAE in Chengdu.  (b) MAE in Xi'an.

**Figure 4: Effect of pre-training in travel time estimation.**

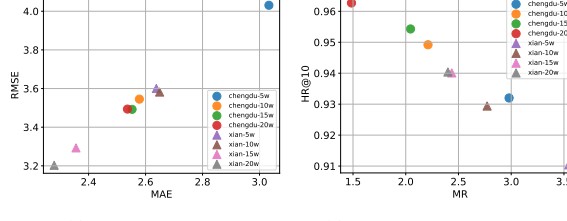

(a) Speed Inference.  (b) Similar Trajectory Search.

**Figure 5: Model Capacity.**

It further demonstrates the potential of JGRM as a large model for transportation infrastructure.

## 6 CONCLUSION

In this work, we design a framework that learns a robust road segment representation and trajectory representation by jointly modeling GPS traces and routing traces. Specifically, we have proposed corresponding encoders for each of the two trajectory characteristics. The GPS encoder uses hierarchical modeling to mitigate noise and redundancy from the GPS trajectory. The route encoder embedded with spatio-temporal information encodes route trajectory with the autocorrelation of sequence. The outputs of the two encoders are fed into the modal interactor for information fusion. Finally, two self-supervised tasks were designed to optimize the model parameters, which are MLM and Match. Extensive experiments on two real-world datasets demonstrated the superiority of JGRM. In the future, we will further explore the JGRM framework for dynamic road segment representation to sense the road state in real time.

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

# 7 APPENDICES

## 7.1 Datasets.

**Table 3: Details of the Datasets**

| Datasets | Chengdu | Xi'an |
|---|---|---|
| Region Sizes ($km^2$) | 68.26 | 65.62 |
| # Nodes | 6450 | 4996 |
| # Edges | 16398 | 11864 |
| # Trajectories | 2140129 | 1289037 |
| Avg. Trajectory Length ($m$) | 2857.81 | 2976.52 |
| Avg. Road Travel Speed ($m/s$) | 11.35 | 9.65 |
| Avg. Trajectory Travel Time ($s$) | 436.12 | 516.24 |
| Time span | 2018/11/01 - 2018/11/15 | |

## 7.2 Experimental settings.

**A. Details of downstream tasks.**

- **Road Classification**: The task distinguishes the types of road segments, similar to node classification in graph mining. In practice, we choose the four most frequently occurring labels (e.g., primary, secondary, tertiary and residential) to evaluate the segment representations, which are sourced from the road network. These labels are used to train classification head that have one linear layer and Softmax activation function. Due to the limited road segment, we use 100-fold cross-validation for the evaluation, setting same as [25]. Classification accuracy was measured using Mi-F1 (Micro-F1) and Ma-F1(Macro-F1).

- **Road Speed Inference**: The task is to estimate the average speed for each road segment, which is a regression problem. The predicted targets are computed from GPS trajectories. Since the average speed distribution is bimodal, we transform the label using the normal distribution transformation. Specifically, the road segment representations are fed into a linear regression head for inference, which outputs the predicted results. Then, the final results are produced by inverse transforming the predicted results. In this task, MAE (Mean Absolute Error) and RMSE (Root Mean Squared Error) are used to evaluate the model performance in 5-fold cross-validation.

- **Travel Time Estimation**: The task takes the route trajectories as input and outputs the regression values to estimate the travel time. To avoid information leakage, we only use route trajectories and route encoder to get the trajectory representations. And the time information in the route trajectory is masked. Given the complexity of the task, we use the multilayer perceptron as the regression head. The activation function is ReLU. Ground truth is normalized during training and inverted during testing. We used MAE (Mean Absolute Error) and RMSE (Root Mean Squared Error) as metrics to evaluate model performance with 5-fold cross-validation.

- **Top-k Similar Trajectory Query**: The task aims to find the trajectory in the database that is most similar to the query trajectory. In preparation, we randomly selected 50k trajectories from the test set as the database. Among them, we randomly selected 5k trajectories as query trajectories. We use the detour strategy to augment the query trajectories to obtain the corresponding key trajectories. The main idea of detour is to ensure that the origin and destination of a route trajectory remain unchanged, and replace the sub-trajectory with another available route deviating from the original one. In this paper, our detour rate is 17.58%. The details of the detour strategy are added in the Appendix. MR (Mean Rank), HR@10 (Hit Ratio@10), and No hit were employed to evaluate the model performance. Among them, MR refers to the average rank of the key trajectory in the returned query results. To minimize the effect of noise, we keep only the first 1k results of each query to calculate this metric. HR@10 is the recall of key trajectories in the top 10 query results. And no hit is the number of key trajectories that do not appear in the top 10 query results. Computational details reference [17]. Since the detour strategy cannot generate GPS trajectories, the trajectory representations in task 4 use only route trajectory and route encoder.

**B. Detour Strategy in Top-k Similar Trajectory Query.**

For each selected query trajectory, we randomly select a subpath of the route. The length of the subpath is $r\%$ of the total length of the route. We extract the beginning and ending segments of the subpath as the origin and destination to perform the reachable route search algorithm on the road network. The generated routes must satisfy that the area enclosed by the new and original routes is greater than the threshold $\lambda_1$. And the routes must not be longer than 1/3 of the query trajectory. This is done to avoid generating trivial solutions.

## 7.3 Additional Experiments.

**A. Model Comparison in Xi'an.**

Similar to result in Chengdu, The experimental results of Xian are shown in Table 4, where the proposed JGRM method performs best on eight metrics on four downstream tasks.

**B. Ablation Experiments in Xi'an.**

Table 5 shows the results of Xi'an's ablation experiments. The conclusion is the same as in Chengdu. It can be seen that the proposed JGRM has the best overall performance compared to other variants. Some of the variants will show better performance on some specific tasks.

**C. Model Transferability Study.**

In Table 6, we present the experiment results results applied to cross-city scenarios to evaluate the model's migratability. Two types of experiments are considered. Zero-shot adaptation refers to using model parameters trained in source city to be directly applied in target city. Few-shot fine-tuning involves using a small amount of data from target city to fine-tune the model trained in source city. In both settings, the road network embedding is randomly initialized on the target city.

Experimental results show that JGRM performs well on section-level tasks, achieving 90% performance of models trained directly on the target city. However, performance on trajectory level tasks are poor. The transferability of the trajectory representation is limited by the fact that people in different cities have different driving habits. The performance in trajectory level tasks were improved as they were fine-tuned on target city. At the same time, the performance of the section-level task deteriorates, which may be due to the fact that only a small number of road segments were observed in the limited data. These road segment representations are updated by the

**Table 4: Model comparison on four downstream tasks in Xi'an.**

|  | Road Classification | | Road Speed Inference | | Travel Time Estimation | | Top-k Similar Trajectory Query | | |
|---|---|---|---|---|---|---|---|---|---|
|  | Mi-F1 | Ma-F1 | MAE | RMSE | MAE | RMSE | MR | HR@10 | No Hit |
| Embedding | 0.4382 | 0.3003 | 3.2619 | 4.1949 | 104.5929 | 137.0655 | 4.0946 | 0.9031 | 0 |
| Word2vec | 0.5962 | 0.5559 | 3.2242 | 4.1103 | 92.9827 | 129.9678 | 5.795 | 0.8617 | 0 |
| Node2vec | 0.4283 | 0.3827 | 3.2945 | 4.236 | 89.6014$^†$ | 122.2406$^†$ | 3.1167$^‡$ | 0.923$^‡$ | 0 |
| GAE | 0.462 | 0.436 | 3.2496 | 4.1794 | 90.2352$^‡$ | 122.9764$^‡$ | 3.5626 | 0.9141 | 0 |
| Traj2vec | 0.5658 | 0.4195 | 2.7798$^†$ | 3.6768$^†$ | 107.8969 | 144.248 | 51.6097 | 0.6221 | 361.5 |
| Toast | 0.7055$^†$ | 0.6606$^†$ | 3.1145 | 4.0025 | 92.9093 | 129.3365 | 5.0072 | 0.869 | 0 |
| PIM | 0.512 | 0.4671 | 3.2367 | 4.1845 | 91.0666 | 123.6043 | 4.243 | 0.8947 | 0 |
| Trember | 0.6627$^‡$ | 0.6212$^‡$ | 3.2052 | 4.1269 | 98.8188 | 134.7582 | 9.5947 | 0.8084 | 0 |
| START | 0.4557 | 0.3298 | 3.2211 | 4.1331 | 105.8333 | 138.6432 | **2.5158** | 0.9283$^†$ | 6.7 |
| JCRLNT | 0.609 | 0.5179 | 3.1651$^‡$ | 4.0864$^‡$ | 100.8771 | 133.8522 | 13.4306 | 0.7659 | 0 |
| JGRM | **0.7823** | **0.7703** | **2.6494** | **3.5818** | **87.166** | **119.2541** | 2.7714$^†$ | **0.9294** | 0 |
| JGRM* | **0.8758*** | **0.8698*** | **2.2029*** | **3.1765*** | **86.2855*** | **118.9211*** | **1.2983*** | **0.9682*** | 0 |
| improvement | 10.89% | 16.61% | 4.92% | 2.65% | 2.79% | 2.5% | / | 0.12% | / |
| improvement* | 24.14% | 31.67% | 26.19% | 15.75% | 3.84% | 2.79% | 93.78% | 4.3% | / |

**Table 5: Ablation experiment on four downstream tasks in Xi'an.**

|  | Road Classification | | Road Speed Inference | | Travel Time Estimation | | Top-k Similar Trajectory Query | | |
|---|---|---|---|---|---|---|---|---|---|
|  | Mi-F1 | Ma-F1 | MAE | RMSE | MAE | RMSE | MR | HR@10 | No Hit |
| JGRM | 0.7823 | 0.7703 | 2.6494 | 3.5818 | 87.166 | 119.2541 | 2.7714 | 0.9294 | 0 |
| w/o MLM Loss | 0.5327 | 0.4128 | 3.2402 | 4.1623 | 115.9861 | 148.8677 | 75.0366 | 0.0768 | 3855.2 |
| w/o Match Loss | 0.7793 | 0.7666 | 2.5667 ↑ | 3.5338 ↑ | 87.3213 | 119.262 | 2.7729 | 0.9319 | 0 |
| w/o GPS Branch | 0.7003 | 0.6869 | 2.7983 | 3.7388 | 87.1901 | 119.3732 | 2.2322 ↑ | 0.9441 ↑ | 0 |
| w/o Route Branch | 0.6248 | 0.5717 | 2.7472 | 3.5753 | 98.0748 | 131.2151 | 5.7801 | 0.8663 | 0 |
| w/o Time Info | 0.7745 | 0.7601 | 2.5816 ↑ | 3.5254 ↑ | 87.5762 | 119.8214 | 5.65 | 0.8655 | 0 |
| w/o Mode Interactor | 0.6268 | 0.5757 | 2.8074 | 3.7472 | 87.2887 | 119.3806 | 2.0412 | 0.9492 ↑ | 0 |
| w/o GAT | 0.7987 ↑ | 0.7846 ↑ | 2.6676 | 3.5982 | 87.2087 | 118.8381 ↑ | 1.7644 ↑ | 0.956 ↑ | 0 |
| w/o Mode Emb | 0.7802 | 0.7691 | 2.4292 ↑ | 3.3746 ↑ | 87.0462 ↑ | 118.757 ↑ | 3.1417 | 0.9245 | 0 |

**Table 6: Model transferability across two citys.**

|  |  | Road Classification | | Travel Time Estimation | |
|---|---|---|---|---|---|
|  |  | Mi-F1 | Ma-F1 | MAE | RMSE |
| Zero Shot | C→X | 0.7252 | 0.6873 | 109.206 | 141.6533 |
| Adaptation | X→C | 0.7295 | 0.6916 | 106.5079 | 139.2584 |
| Few Shot | C→X | 0.6712 | 0.6662 | 105.2994 | 134.9308 |
| Finetune | X→C | 0.6802 | 0.6779 | 99.1057 | 128.7578 |

C and X in the table are abbreviations for Chengdu and Xi'an.

observed data; they are dynamic representations at a given moment in time, rather than the static representations we would expect. We believe that JGRM can achieve more consistent performance as the training data on the target city is collected.

**D. Parameter Sensitivity.**

We further conduct the parameter sensitivity analysis for critical hyperparameters, including embedding size $d$, route encoder layers $L1$, mode interact layers $L2$, the length $l$ and probability $p$ of mask. The encoding size experiments are shown in Figure 6, and the results show that the larger the encoding size, the better the performance. The best results occur at an encoding size of 1024, suggesting that

there are complex patterns in the trajectory that need to be carried by a higher dimensional representation space.

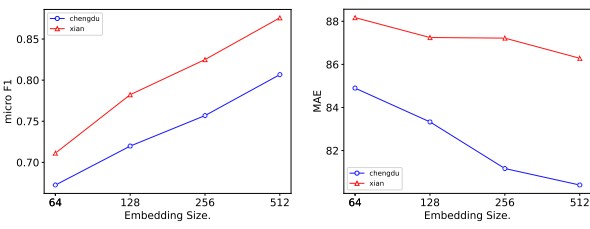

(a) Micro F1 in Road Classification.    (b) MAE in Travel Time Estimation.

**Figure 6: Different # of Embedding Sizes.**

Figure 7 illustrates the parameter sensitivity of the route encoder. The results show that this parameter performs differently in different cities, but overall the best results are obtained with a value of 2. This may be due to the different complexity of trajectories in different cities.

The experimental results of the modal interactors are shown in Figure 8. Modal interactor with 2 layers performs best

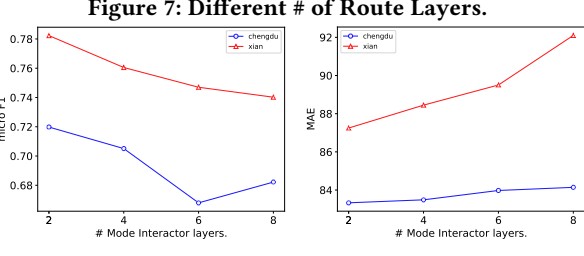

(a) Micro F1 in Road Classification.  (b) MAE in Travel Time Estimation.

**Figure 7: Different # of Route Layers.**

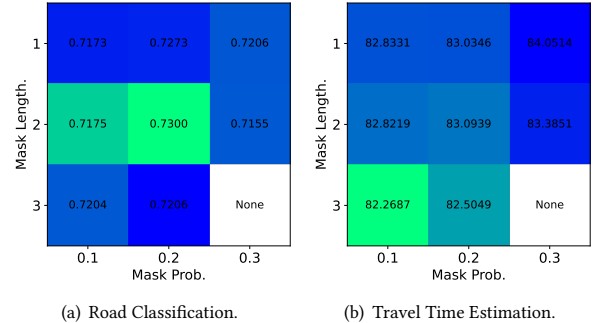

(a) Micro F1 in Road Classification.  (b) MAE in Time Estimation.

**Figure 8: Different # of Mode Interact Layers.**

Finally, we compared different combinations of mask length and mask probability on the Chengdu dataset. Overall, the model performs best when about 40% of the trajectory is masked. And, it turns out that the longer mask length improves the model's performance on trajectory-level tasks when the same number of tokens is used for the mask.

(a) Road Classification.  (b) Travel Time Estimation.

**Figure 9: Different # of Mask Settings.**

**E. Case Study.**

We randomly select three trajectories from the Chengdu dataset and use our JGRM and suboptimal Node2vec to obtain trajectory representations for top-k similar trajectory query, respectively. The

results are presented in Figure 10. where the three columns represent the results of the top-1, top-3, and top-10, respectively, with the odd-numbered rows referring to the results of JGRM and the even-numbered rows referring to the results of Node2vec. Red indicates query trajectory, green indicates key trajectory, and blue indicates query results. The results show that while the graph embedding-based approach is sensitive to changes in road segments, it is unable to capture sequential, temporal and kinematic information. This means that Node2vec can't distinguish between two trajectories that are opposite to each other, nor can it distinguish between trajectories from different users at different times under the same OD. In contrast, our method is more sensitive to detour behavior and is able to capture subtle changes in trajectories.

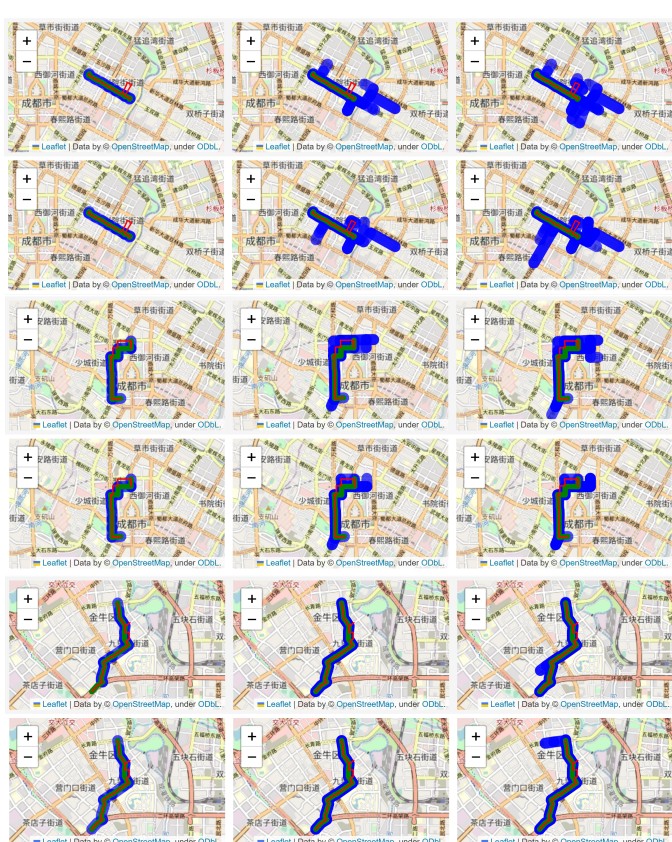

**Figure 10: Case Study in Chengdu.**

