# OpenReview forum: "More Than Routing: Joint GPS and Route Modeling for Refine Trajectory Representation Learning"
_ACM.org/TheWebConf/2024/Conference — TheWebConf24_

### Official Review · Reviewer_JHsy · 2023-11-11

**Novelty:** 5
**Technical Quality:** 5

**Review:**

## Strengths

- The paper proposes joint modeling of GPS and route trajectories for trajectory representation learning, complemented by well-designed self-supervised tasks, demonstrating potential generalization to future research.
- The writing is clear and logical. The authors thoroughly explain the limitations of prior methods and articulate how their approach addresses these issues.
- Comprehensive experiments on real-world datasets are well-presented, and the analysis of the experimental results is detailed and insightful.

## Weaknesses

- Abbreviations like "GAL," "TE," "MLM," "Match," and "CMM" lack full names when introduced in the Introduction section, requiring clarification.
- In section 4.1, when stating, "modeling GPS trajectories as traditional sequence data would... accommodate long trajectories," specific references supporting this claim could be beneficial.
- In Formula (6) of section 4.2, what does "FFN" stand for? Additionally, clarification is needed for the terms “{GE}^{’}” and “\mathrm{\Delta}t_{j}” in the formula.
- How were the hyperparameters “w_1, w_2, and w_3” in Formula (17) for balancing the three tasks determined? Could you elaborate on their specific values or the methodology used to set them in the experimental settings? Exploration of the impact of these hyperparameters on results would add value to the experiments.
- Could you explain the meaning of the dotted red line in Figure 4? Its purpose and significance are unclear and require clarification.
- Some editing errors are noted. In Section 5.3, the results of ablation experiments in Chengdu should be shown in Table 2 instead of Table 5. Additionally, in the CONCLUSION section, "spatio-temporal" should be corrected to "spatial-temporal.”

**Questions:**

As above

**Reviewer Confidence:**

2: The reviewer is willing to defend the evaluation, but it is likely that the reviewer did not understand parts of the paper

**Scope:**

2: The connection to the Web is incidental, e.g., use of Web data or API

---

### Official Review · Reviewer_JCdU · 2023-11-18

**Novelty:** 5
**Technical Quality:** 5

**Review:**

The authors propose a novel representation learning framework called JGRM, which incorporates GPS and route trajectories. Extensive experiments on two real-world datasets validate the effectiveness of the proposed framework.

Strength:
- JGRM integrates GPS trajectory and route information, which has not been explored extensively in previous work.

- The proposed approach outperforms several variants and achieves state-of-the-art performance on multiple tasks.

Weakness:

- The authors use intra-road and inter-road BiGRUs to encode segment entities and refine the segment representations. However, it is unclear how the network structures of these two BiGRUs differ. The authors should provide a more detailed explanation of the differences between these two BiGRUs. Additionally, it is well-known that GRUs have difficulty capturing long-term dependencies in time series data, while transformers are better suited for this task. The authors have used transformers in the model interaction process, but it is unclear why they did not use them in the intra-road and inter-road BiGRUs. The authors should explain more about why they chose to use BiGRUs instead of transformers in these model components.


- The authors extract 7 features for each GPS trajectory, including longitude, latitude, speed, acceleration, angle delta, time delta, and distance. However, it is unclear what criteria are used to select these features. The authors should provide a more detailed explanation of the rationale behind the feature selection process. Furthermore, it appears that these features have different data properties, and it is unclear how they were encoded. The authors should provide a more detailed explanation of how each feature is encoded and how these features affect the performance of the model. Similar questions arise for the route trajectories.

- Several terms used in the paper lack explanation, such as GATLayer, ME, PE, TransEncoder, and FFN. The authors should provide a more detailed description of the terms.

- The authors evaluate the performance of their method on four downstream tasks. However, in the analysis of the pre-training effect, they only considered the impact on travel time estimation. It is unclear whether pre-training would have similar effects on other tasks. The authors should conduct a more comprehensive analysis of the pre-training impact on all four downstream tasks.

- The authors construct 8 variants for ablation experiments to evaluate the effects of each module in JGRM. However, this only demonstrates the effectiveness of each module individually. It does not indicate whether the modules in JGRM are separable or replaceable. For example, it is unclear why GRU is used instead of Transformer, which may perform better. The authors should conduct a module replacement analysis to understand the potential for alternative architectures.

- Although the authors conduct extensive experiments on multiple tasks to demonstrate the effectiveness of the proposed method, I would like to see more analysis of the relationships between representations. For example, what happens when a representation of a trajectory from point X to point Y is added to a representation of a trajectory from point Y to point X? What is the relationship between the sum of representations A and B and the representation of C?

- The authors should provide further clarifications on the effect of pre-training on travel time estimation. Specifically, the authors should explain why their method exhibits different trends on the Chengdu and Xi'an datasets despite the same sample size.

- There are many grammar errors and sentence repetitions in the paper. The authors should carefully proofread the paper to correct these issues. Specifically, the following areas require attention:
  (1)More Than Routing: Joint GPS and Route Modeling for Refine Trajectory Representation Learning
  (2)To fill this gap, we propose a novel representation learning framework that Joint GPS and Route Modelling based on self-supervised technology
  (3)Note that We use only the one that most closely resembles the current query trajectory as the negative sample in each retrieval.
  (4)A hierarchical bidirectional GRU … this two-stage modeling …A hierarchical bidirectional GRU … this two-stage modeling

**Questions:**

N/A

**Reviewer Confidence:**

3: The reviewer is confident but not certain that the evaluation is correct

**Scope:**

4: The work is relevant to the Web and to the track, and is of broad interest to the community

---

### Official Review · Reviewer_VCW4 · 2023-11-21

**Novelty:** 3
**Technical Quality:** 5

**Review:**

Summary
This paper focuses on trajectory representation learning by combing GPS trajectory and route trajectory. They propose a model named JGRM, which contains the GPS encoder to model the characteristics of road entities, the route encoder that considers the spatio-temporal correlation of trajectories, and the modal interactor for information fusion. They conduct extensive experiments such as four downstream tasks on two real-world datasets validate the effectiveness of JGRM.

Strength
1.	The structure of the paper is clear and easy to understand.
2.	This paper conducts various experiments based on real-world data and provides detailed results.

Weakness
1.	The motivation of combing GPS trajectory and route trajectory is not clear.
2.	The challenges mentioned in the work appear to be extensively studied problems. It is not clear what new aspects this work tries to arise.
3.	The design should be better motivated to correspond to the challenges and goal.
4.	The motivation and setting of some downstream tasks are not clear.
5.	More explanations are needed for the experimental results.


Detail comments:
1.	What specific information in GPS helps trajectory representation learning compared to the route? The authors provide some arguments, such as movement details and interactions of objects with the space, but these points appear vague. The reviewer suggests that the authors provide quantitative results to support these arguments.
2.	The idea of fusing GPS and route is similar to [5]. The authors claimed at Line 201 that 'While Toast [5] ... but does not model them well in a fused way.' However, it appears that [5] did fuse them based on an aggregated loss function.
3.	The author clarifies that 'The raw GPS trajectories are mapped into the road network using the map matching algorithm to obtain the route trajectories.' Does this imply that GPS trajectories already contain all the information of the routes? If so, why is there still a need to fuse the routes with GPS trajectories?
4.	The motivation and setting of some downstream tasks are not clear. For example, why do we need to estimate the travel time if we already have complete trajectories with time as the model input? The authors tune this issue by masking time information and only using route trajectories. If so, how does this help validate the effectiveness of fusing two types of trajectories? A similar question arises regarding road speed inference. Additionally, what serves as the ground truth for these tasks, such as speed inference? If these ground truth can also be calculated directly from the trajectories, why do we still need representation learning?
5.	The author does not specify the granularity of the road network, such as the length of the roads. Will this granularity affect the results?
6.	In ablation study, some variants outperform the proposed method. Does that mean some components are not necessary? It needs more explanations.
7.	Some words or definition should be better explained, such as L106: “direct self-observation”.

**Questions:**

1. Quantitative support for fusing GPS and routes.
2. How the challenges mentioned in this work is different from others.
3. The motivation and setting of the downstream tasks.
4. Experimental results explanation.
5. How the design corresponds to addressing the challenges.

See details of the questions in the review.

**Reviewer Confidence:**

4: The reviewer is certain that the evaluation is correct and very familiar with the relevant literature

**Scope:**

3: The work is somewhat relevant to the Web and to the track, and is of narrow interest to a sub-community

---

### Official Review · Reviewer_Ek6q · 2023-11-26

**Novelty:** 5
**Technical Quality:** 4

**Review:**

The authors introduce JGRM, a novel representation learning framework for GPS trajectories, focusing on joint GPS and route modeling using self-supervised technology. The framework uses two specialized encoders to capture representations of route and GPS trajectories, integrating them through a shared transformer for inter-modal information interaction. JGRM addresses challenges like GPS trajectory uncertainty, spatio-temporal correlation in route trajectory, and complexity in information fusion. The approach demonstrates improved performance in road segment and trajectory representation tasks across two real-world datasets, setting a new standard for trajectory representation learning.

Overall, this paper is easy to understand, although with some typos. And In Section 2.1, there should be many space characters between different sentences. In line 244-245, is it a unfinished sentence? The Fig. 2 should be more abundant with more information.

Does the TransEncoder in Eq. 7 and 10 represent  the same one with same parameter?

The parameters described in this paper are not always introduced clearly, which makes it hard to reproduce.

**Questions:**

In Section 2.1, there should be many space characters between different sentences. In line 244-245, is it a unfinished sentence? The Fig. 2 should be more abundant with more information.

Does the TransEncoder in Eq. 7 and 10 represent  the same one with same parameter?

The parameters described in this paper are not always introduced clearly, which makes it hard to reproduce.

**Reviewer Confidence:**

3: The reviewer is confident but not certain that the evaluation is correct

**Scope:**

3: The work is somewhat relevant to the Web and to the track, and is of narrow interest to a sub-community

---

### Decision · Program_Chairs · 2024-01-22

**Decision:**

Accept

**Comment:**

The paper proposes joint modeling of GPS and route trajectories for trajectory representation learning, which can be meaningful for many downstream tasks. Although the paper is generally well-written and easy-to-understand, it still needs to be revised extensively.

 - Clearly states the motivation of combing GPS trajectory and route trajectory, as well as the algorithm design.
 - More explanations are needed for the experimental results.
 - More clarification on the choice of downstream tasks.
 - The relationship to Web is low.